# Risk Mapping of Water Supply and Sanitary Sewage Systems in a City in the Brazilian Semi-Arid Region Using GIS-MCDA

**Marcelo Carlos de Oliveira Silva \*, Rochele Sheila Vasconcelos**  **and José Almir Cirilo**

Agreste Academic Center, Federal University of Pernambuco, Caruaru 55002-917, PE, Brazil
\* Correspondence: marcelo.carlosoliveira@ufpe.br

**Abstract:** It is essential to visualize water scarcity as a result of an inappropriate process of appropriation and the use of natural resources. This understanding has been gaining more and more prominence in studies in Brazil and also in the world. In this context, this research aims to map the risk of water shortages and sewage insufficiency in a Brazilian semi-arid city using GIS-MCDA. The secondary data used in this research were collected from IBGE, Compesa, and the City Hall of Caruaru and were processed using the QGIS 3.12 Bucareşti software. The Pernambuco Tridimensional database and the Analytical Hierarchy Process method were used in the process to generate the maps. After collecting and analyzing the data, it was seen that, according to the water shortage risk map, seven neighborhoods had a "Very high" risk, in which the criteria with the greatest weight were the distance from distribution reservoirs, the main supply network and altimetry. The map of the degree of sanitary sewage insufficiency showed that four neighborhoods have a "Very high" degree; these neighborhoods are far from the main sewage network and from sewage treatment stations and have the lowest rates of households served by the system. Such characteristics need to be highlighted in the planning and implementation of water and sewage services. Thus, it is concluded that the use of high-resolution spatial databases for the planning of urban services, as carried out in the present work, provides a greater level of confidence for solutions that can be implemented in the expansion of service networks to the population.

**Keywords:** AHP; smart cities; water shortage; urban planning



## 1. Introduction

According to [1,2], basic sanitation has become a central topic in discussions about smart cities and is an increasingly urgent concern in academia and public management, mainly due to the deterioration of the quality of water sources and water scarcity.

According to [3], since the 1980s, water use has increased across the planet at a rate close to 1% per year as a consequence of greater socioeconomic development, population growth, and changes in consumption patterns. This demand is expected to increase at a similar rate until 2050, which will correspond to an increase between 20% and 30% in relation to the current value of consumption. In this context, for [4], when natural resources are misused, soil desertification advances for a lack of water, water sources become more insufficient and poverty and the exodus to larger cities grow.

In addition to the increase in the demand for water, it is also worth mentioning the population projections that, according to [5], the global population is expected to reach 8.5 billion in 2030, 9.7 billion in 2050 and 10.9 billion in 2100. In 1950, 30% of the world's people lived in urban areas. Since then, in 2018, the urban population was already 55%, and by 2050, the projection is that 68% of the world's population will live in cities. These projections have pushed water supply and sewage systems to their limits. When talking about semi-arid regions, this scenario is even more alarming due to the recurrent water crises in such regions and the absence or lack of adequate infrastructure in many locations.

As the world continues in its urbanization process, researchers such as [6] highlight the relevance of sustainable development, which increasingly needs an efficient management of urban growth, especially in developing countries, where there is fast-paced and disorganized urbanization. Thus, for [7], investments in integrated policies are needed to improve the lives of residents in both urban and rural areas, developing their economic, social and environmental ties and providing greater well-being for the population.

According to [8], numerous factors such as climate, population density and economic and environmental problems have led cities to water crises and rationing. Considering the Brazilian Northeast, between 1559 and 2011, there were 72 droughts—an average of one drought every 6.3 years—according to over 452 years of records on this phenomenon [9]. As highlighted by [10], the climatic conditions and the high population density of the Brazilian semi-arid region set limits on the water potential of this region, which, according to [11], is poor in terms of the volume of runoff of water from rivers. This situation can be explained by the temporal variability of rainfall and the dominant geological characteristics, where there is a predominance of shallow soils over rocks. Thus, water management in the northeastern basins is overloaded in relation to other basins in the country due to the requirements of the construction, operation and maintenance of water infrastructure that ensures the availability of water.

The significant water rationing in the city of Caruaru-PE, driven mainly by the lack of rain and the deficiency in water distribution, should point out new reflections and actions in the management of urban waters. What is desired in the near future is an integration between conventional and alternative water supply systems, where rainwater and gray water—these, according to [6], come from washbasins, showers and washing machines—will play a considerable role in providing new guidelines for urban water management. Despite this, the state of Pernambuco has a large part of its territory inserted in the semi-arid region, in areas of significant water scarcity, of which Caruaru is a part, being an environment of expressive complexity for the analysis of alternatives for water supply.

According to [12], 2319 municipalities (41.8%) registered water supply rationing in the 12 months prior to the Municipal Basic Information Survey (MUNIC)—2017. Added together, these municipalities represented 49.2% of the Brazilian population. In addition, according to [13], only five Brazilian municipalities have a sewage collection network throughout urban and rural areas. In only 11 of the 27 units of the federation in Brazil (26 states plus the federal district) did the municipalities have more than 50% of the sewage collection network. In 2017, there were 34.1 million households without sewage service in the country, of which 13.6 million were in the Northeast, a region in which only 16.2% of the municipalities had this service.

In this scenario, according to [14], analyzing socio-environmental impacts in urban space becomes fundamental for the planning, development and ordering of cities. The growth of society in the urban environment requires the adequate use of its natural resources such that that they are not exhausted. Therefore, the Geographic Information System based the Multi-Criteria Decision Analysis (GIS-MCDA) as the preferred approach in several studies because it involves a combination of multiple criteria in a weighted way and also produces visual results, which is important for decisions in the urban environment, being used to: (i) delineate MAR (Managed Aquifer Recharge) sites [15,16]; (ii) identify potential zones for rainwater harvesting in a semi-arid area [17]; (iii) select priority management programs for watersheds [18]; (iv) assess the risk of urban flooding [19,20]; (v) decide the multi-criteria support for water management [21]; (vi) optimize the layout of water supply pipeline systems [22]; (vii) carry out mapping of water scarcity risks [23]; (viii) create zoning of areas at risk of deforestation [24]; (ix) identify water distribution points [25]; (x) analyze the feasibility of supply systems with groundwater abstraction [26]; (xi) identify Concentrated Solar Power (CSP) and Photovoltaics (PV) points to assess the potential of solar energy [27]; (xii) identify potential areas for marina construction [28]; (xiii) select landfill sites using GIS-MCDA [29].

According to [30], the use of multicriteria analysis methods has been increasingly present in scientific works; today, it is possible to find several methods of comparison between pairs, including: ANP (Analytic Network Process), MACBETH (Measuring Attractiveness by a Categorical-Based Evaluation Technique) and AHP (Analytical Hierarchy Process), among others. Of these, the method used in this work was the AHP, due to it being one of the first decision-making methods with several criteria and one of the most common methods in academia.

From this, the discussion of this research is inserted in the scope of the search for smarter cities, which promote adequate and tangible water supply systems and sanitary sewer systems (SSS) to the entire population through management and urban planning actions. Such a study is necessary due to the periodic water crises and the insufficiency of the sewage collection network in the area under study. The Brazilian semi-arid region experiences a situation of water scarcity due to the uneven distribution of rainfall, which, associated with low investments in infrastructure and the poor management of water resources, highlights the lack of conventional systems for other sources of water supply.

According to [31], to meet the future demands of water and sanitation, amid the challenges of aging conventional infrastructure, population growth and climate change, a series of possibilities, new strategies and innovative management will be needed. In this context, this research aims to map the risk of water shortages and sewage insufficiency in a Brazilian semi-arid city using GIS-MCDA.

## 2. Materials and Methods

### 2.1. Study Area

According to [9], the Brazilian semi-arid region is one of the rainiest compared to other semi-arid regions around the world; it has an average annual rainfall of 750 mm. On the other hand, it has an average evapotranspiration potential of about 2500 mm/year, in addition to having a shallow soil with low infiltration and storage capacity, contributing to water scarcity. The municipality of Caruaru is located in the Agreste mesoregion inserted in this semi-arid region, 130.07 km away from the state capital, Recife. The seat of the municipality has an altitude of 533.5 m. According to [32], the municipality, with its 933 km$^2$ area, corresponds to 11.31% of the area of the Ipojuca River Watershed.

According to [33], the rainy season begins in February with pre-season rains and lasts until the end of August or into the first half of September. It is worth noting that the rainiest months are May, June and July, and the driest months are October, November and December. Figure 1 shows the location of the municipality of Caruaru and its headquarters.

Regarding its urban area, according to the Caruaru database, the city has 43 neighborhoods (Figure 2 and Table 1). In this study, this database was considered for the analysis of the SUAA and SES. The population applied by neighborhoods was found using the data from the census sectors that make up each neighborhood and the IBGE population projection methodology [34]. Between 2010 and 2020, the neighborhoods of Boa Vista, Indianópolis, Nova Caruaru, Petrópolis, Rendeiras, Salgado and Universitário were subdivided. This subdivision generated 12 new neighborhoods. In addition to these, eight additional new neighborhoods emerged in other areas during the same period, going from 23 in 2010 to 43 in 2021, evidencing the accelerated expansion of the urban area of Caruaru, where, in many cases, the public power is unable to keep up with it to provide urban infrastructure systems that satisfactorily meet the needs of the population.

With regard to the holders of water services, in Caruaru, Companhia Pernambucana de Saneamento (Compesa) is responsible for the collection, transport and supply of water, as well as the collection and treatment of sewage generated in the city. The City Hall of Caruaru (CHC) also performs this service through the Secretariat of Urban Infrastructure and Works. The management of micro and macro drainage systems, in turn, is the responsibility of the city hall. According to [35], the Caruaru supply system has 90,258 connections, serving 105,182 economies and 41,688 sewage connections. In 2017, according to the [36], in Caruaru, the number of active residential-supplied economies was 121,806. In 2020, the city received

water from the Prata Dam, built in 1998 and located in the city of Bonito. The spring has the capacity to store 42 million m³ of water and, in addition to Caruaru, supplies Agrestina, Altinho, Ibirajuba and Cachoeirinha.

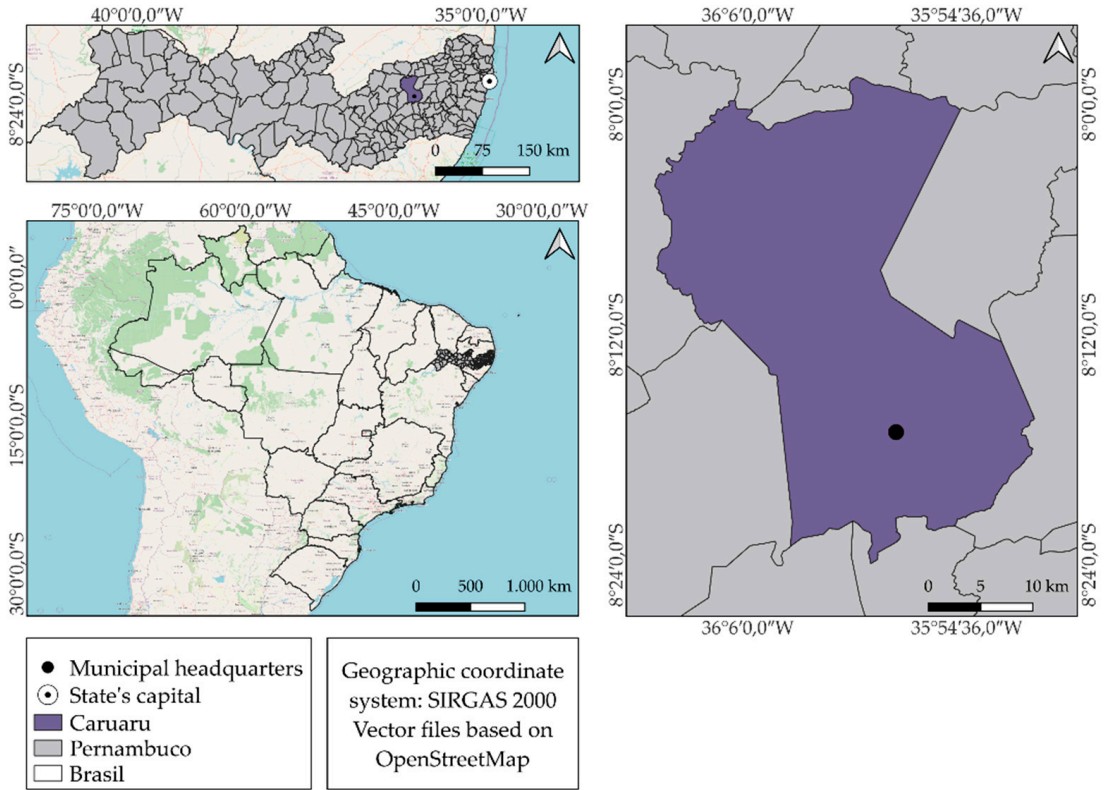

**Figure 1.** Location map of the municipality of Caruaru.

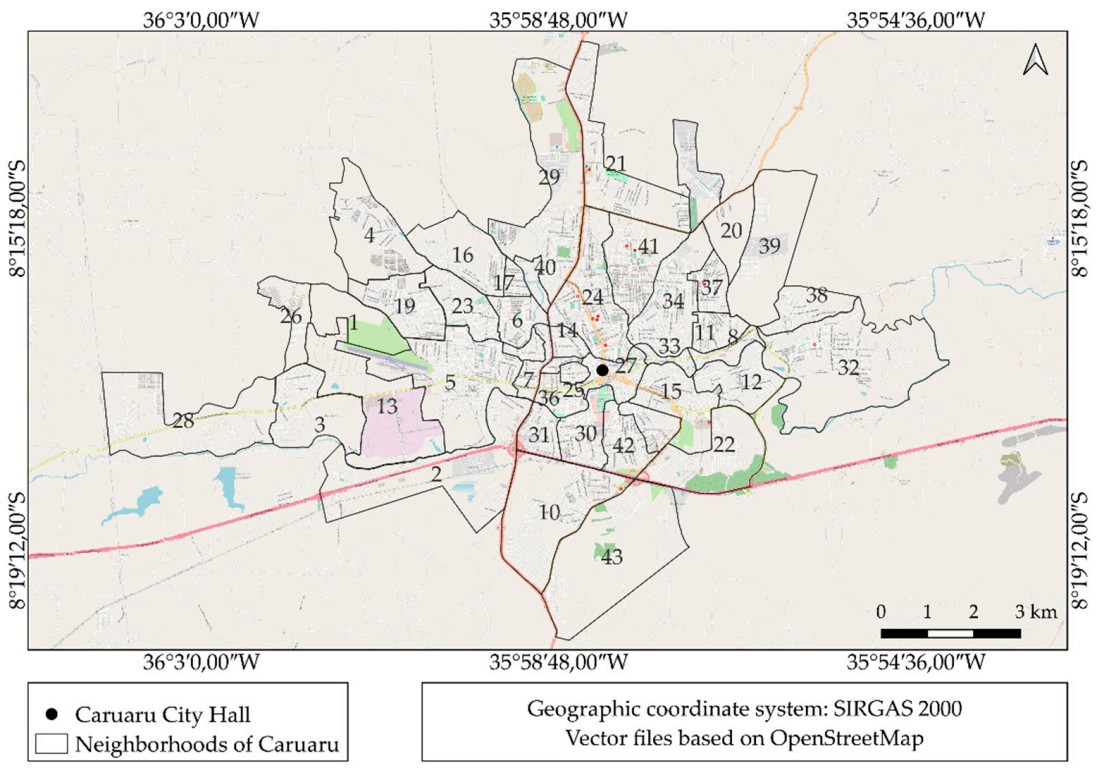

**Figure 2.** Division of neighborhoods in Caruaru.

**Table 1.** Nomenclature of the neighborhoods of Caruaru.

| Index | Name | Index | Name |
|---|---|---|---|
| 1 | Aeroporto | 23 | Maria Auxiliadora |
| 2 | Agamenon Magalhães | 24 | Maurício de Nassau |
| 3 | Alto do Moura | 25 | Morro do Bom Jesus |
| 4 | Andorinha | 26 | Nina Liberato |
| 5 | Kennedy | 27 | Nossa Senhora das Dores |
| 6 | Boa Vista | 28 | Nossa Senhora das Graças |
| 7 | Caiucá | 29 | Nova Caruaru |
| 8 | Cedro | 30 | Petrópolis |
| 9 | Centenário | 31 | Pinheirópolis |
| 10 | Cidade Alta | 32 | Rendeiras |
| 11 | Cidade Jardim | 33 | Riachão |
| 12 | Dep. José Antônio Liberato | 34 | Salgado |
| 13 | Distrito Industrial | 35 | Santa Rosa |
| 14 | Divinópolis | 36 | São Francisco |
| 15 | Indianópolis | 37 | São João da Escócia |
| 16 | Jardim Boa Vista | 38 | São José |
| 17 | Jardim Panorama | 39 | Serras do Vale |
| 18 | João Mota | 40 | Severino Afonso |
| 19 | José Carlos de Oliveira | 41 | Universitário |
| 20 | Lagoa do Algodão | 42 | Vassoural |
| 21 | Luiz Gonzaga | 43 | Verde |
| 22 | Manoel Bezerra Lopes | | |

Also in 2019, it received a supplement from the Pirangi River, with a catchment located in Catende. This began to serve Caruaru, since the Jucazinho reservoir, which was the main source that supplied the city [37], reached dead volume at the end of 2015. The construction of the Pirangi System aims to increase the supply of water to the municipalities supplied by the source of the Prata, in addition to the districts of these municipalities. "The increase will be 300 L per second for Caruaru, which will make it possible to leave the rotation and ensure the safety of the source of Prata" [38]. According to [39], the Guilherme Azevedo, Jaime Nejaim and Serra dos Cavalos dams also supply the municipality of Caruaru. Regarding water consumption, according to [40], the average per capita consumption in 2018 was 83.04 L/inhab/day, a value considered low in relation to the consumptions of other cities with similar characteristics, such as Campina Grande /PB, Mossoró/RN and Patos/PB, as highlighted by [41].

### 2.2. Characterization of the UWSS and the SSS of Caruaru

For an adequate management of sanitation services, according to [42], it is essential to initially carry out a diagnosis of the situation in order to know the strengths and weaknesses of the services that are offered to the population, especially in urban environments of accelerated and disorderly growth, where the public power often cannot keep up with the speed of transformation of these spaces. The secondary data used in this work were collected from IBGE, Compesa, PE3D and the City Hall of Caruaru. Data collection was carried out through the Access to Information Law (Law n. 12,527, of Nov. 18, 2011), going directly to the agencies, or through the internet.

Through the secondary data obtained, the distributed and macro-measured volume for the population of the urban area of the municipality in 2021 was 20,611,914 m$^3$, equivalent to an average of 156 L of water per inhabitant per day, a value 88% higher than the value published by [40]. The latter should reflect the rationing that the population of Caruaru has actually had since 2015. The macro-measured volume of water is treated by the water treatment plants located in the Salgado neighborhoods (34) and Petrópolis (30) and distributed to the supply network through 15 reservoirs located in the urban network, as shown in Figure 3.

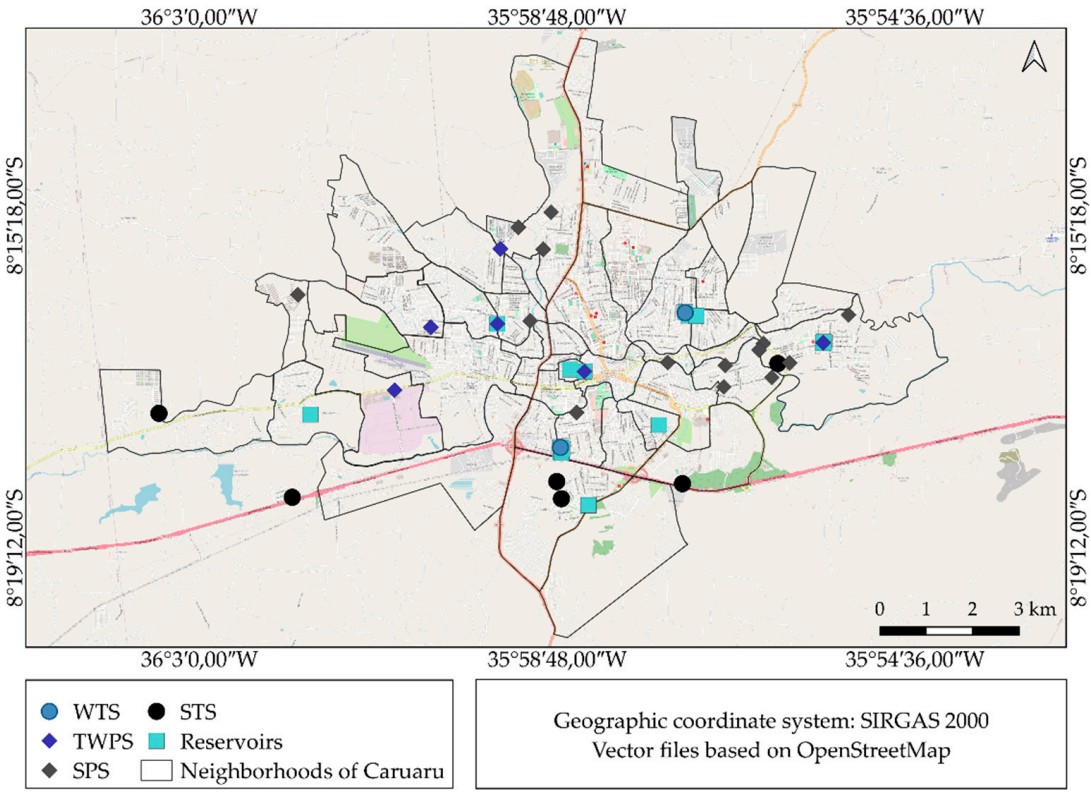

**Figure 3.** Components of the UWSS and the SSS of Caruaru.

Eight (8) TWPSs (treated water pumping stations) also make up the Caruaru water supply system, namely, TWPS Petrópolis (Rua Boa Viagem, Petrópolis), TWPS Salgado (Rua Bartolomeu de Anacleto, Salgado), TWPS Morro Bom Jesus (Rua Cristóvão Colombo, Centro), TWPS S Boa Vista (Rua Santa Maria da Boa Vista, Boa Vista), TWPS Rendeiras (Rua COHAB III, Rendeiras), TWPS Mutirão (José Carlos de Oliveira), TWPS Xique-Xique (Jardim Panorama) and TWPS Alto do Moura (Alto do Moura). It is observed that the areas further north and west of the city have a greater lack of elements that constitute the supply system, which can increase the risks of shortages in these areas.

Regarding the sanitary sewage network, according to [36], the city of Caruaru has 81.3% of households with adequate sanitary sewage. When compared with the other municipalities in the state, it is in the 5th position out of 185; in relation to other cities in Brazil, its position is 896th out of 5570 municipalities. However, according to [40], only 60.95% of the urban population is served by the sewage network. Regarding the annual volume of sewage produced, 80% of the volume of water consumed is considered; that is, 16,489,531 m$^3$ of sewage is produced per year in Caruaru, and the percentage of collection is approximately 45% of the total water connections. The collected sewage is transported and treated in six STSs (sewage treatment stations) in operation, which will receive the reinforcement of another four that are under construction/pre-operation. Figure 3 shows the location of the STS in Caruaru.

The Caruaru sewage system is also comprised of 15 SPSs (sewage pumping stations) in operation (Figure 3) and 13 SPSs in the construction/pre-operation phase. It is observed that the areas located in the North, Northeast and Midwest of the city are more devoid of the components of the sanitary sewage system. As a result, in the first visual analysis, it is inferred that these areas are more susceptible to insufficient adequate sanitary sewage, being areas suitable for receiving programs and projects aimed at mitigating possible urban problems due to the absence of elements that make up the sewage collection and treatment networks.

*2.3. GIS-MCDA for the Proposed Objective*

The criteria considered for mapping areas at risk of water shortages (Table 2) come from [23], with the addition of one more criterion (distance to water treatment plants). These criteria were selected during meetings with stakeholders based on their relevance to the UWSS, as well as the availability of information from institutions and local surveys. They are: population; altimetry; distance to reservoirs; distance from treated water pumping stations; distance from the main network; households served; average monthly income; the distance to water treatment plants.

**Table 2.** Criteria for determining water shortage risks in an urban area.

| Criteria | Description | Data Source |
|---|---|---|
| Criterion 1: Resident population | The larger the population, the greater the potential demand for water; therefore, the greater the risk of shortages. | Census data [43]. 2021 estimate by IBGE's geometric growth rate. |
| Criterion 2: Altimetry | The higher the altitude value, the greater the risk of shortages. | Digital Terrain Model [44]. |
| Criterion 3: Distances to reservoirs | The further away from the reservoir, the greater the risk of shortages. | Calculation of Euclidean distances [39]. |
| Criterion 4: Distances to TWPS | The further away from the TWPS, the greater the risk of shortages. | Calculation of Euclidean distances of the TWPS [39]. |
| Criterion 5: Distances to the main water supply network | The further away from the main network, the greater the risk of shortages. | Calculation of Euclidean distances of the main network (diameters $\geq$ 150 mm) [39]. |
| Criterion 6: Number of households served by the network | The greater the number of households served by the network, the greater the demand and the greater the risk of shortages. | Census data [43]. |
| Criterion 7: Income | The higher the income, the lower the risk of shortages. | Census data [43]. |
| Criterion 8: Distances to WTS | The further away from the ETA, the greater the risk of shortages. | Calculation of Euclidean distances of the WTS [39]. |

The criteria considered for the mapping of areas at risk of insufficient sanitary sewage were: population; altimetry; distance to sewage treatment plants; distance to sewage pumping stations; distance from the main network; households served by the sewage; income network. Such criteria were selected based on the similarity with the criteria for assessing the risk of water shortages and on the expertise of specialists in the sanitation area.

**3. Data Processing**

The shapefiles and raster files used in this study and the operations performed were processed using QGIS 3.12 Bucareşti, which is a free software and an official project of OSGeo (Open Source Geospatial Foundation), a non-profit organization founded in 2006. It is worth noting that shapefile should be understood as a file with vector representation. It is a type of file that has an associated coordinate pair, an attribute table and a visible feature that can be a point, line or polygon [45–47].

The raster, in turn, according to [45], is a matrix representation that consists of the use of a regular checkered grid on which the element being represented is built, cell by cell. A code referring to the studied attribute is assigned to each cell so that the computer knows to which element a particular cell belongs. Given the objective to be achieved and the criteria determined, it is possible to apply the Multicriteria Analysis to solve the research problem.

The criteria must be normalized to allow for the combination between them and must have their weights defined. Thus, the following steps cover: normalization, weighting and combination of criteria.

*Standardization, Weighting and Combination of Criteria*

As highlighted by [48], the various criteria adopted in the decision-making process to measure an alternative may be exposed at different scales or units. Thus, before performing algebraic map operations, an adaptation is necessary, making it possible to match the criteria used. According to [49], the normalized scale is essential in any multicriteria method that needs to integrate measures of comparison in initially different scales.

Data on population, altimetry, distances and/or households served by the network and income were divided into five classes and, according to [50], normalized in a scalar manner by a gradual transition ranging from 1 to 5, unlike the Boolean or binary form, where the condition is a sudden transition with only two possibilities, 0 or 1. For standardization, the r.reclass tool was used, which is applied to reclassify the values of a raster from a text document and is present in GRASS, a module found in QGIS 3.12 Bucareşti (and in other versions as well) used for managing and analyzing geospatial data, image processing and visualization and spatial modeling, among other functions. Then, in Figures 4 and 5, we have the standardized criteria for analyzing the risk of water shortages and the degree of insufficiency of sanitary sewage, respectively.

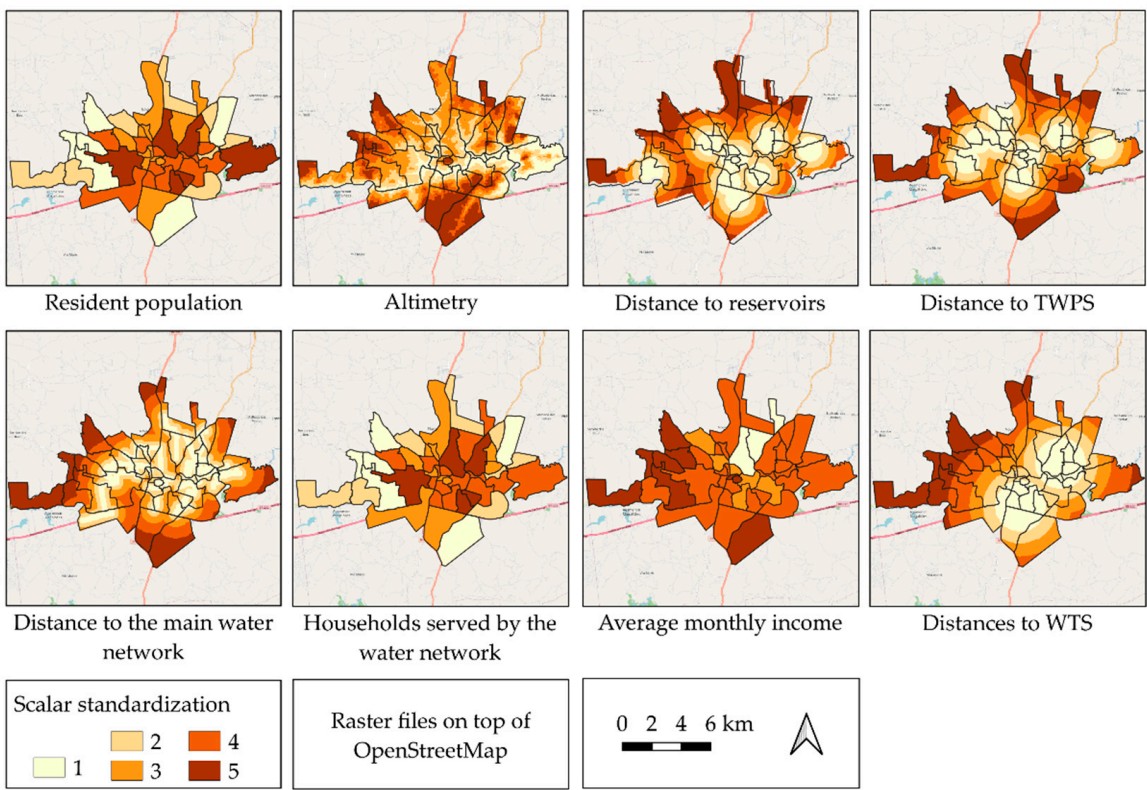

**Figure 4.** Standardization of criteria to assess the risk of water shortages, using a scale of 1 to 5.

The AHP is based on a mathematical structure of n x n matrices; the rows and columns correspond to the n criteria considered for the problem. The value aij represents the relative importance of the criterion in row i against the criterion in column j. The consistency ratio (CR) needs to be evaluated, and, according to [51], it must have a value lower than 0.1 so the values are consistent and the weights can be used. Thus, for [52], the comparison matrices will be constructed by assigning judgments comparing, pair by pair, each element and forming the weights. Table 3, from [53], shows the scale for comparing the criteria according to the degree of importance, pair by pair.

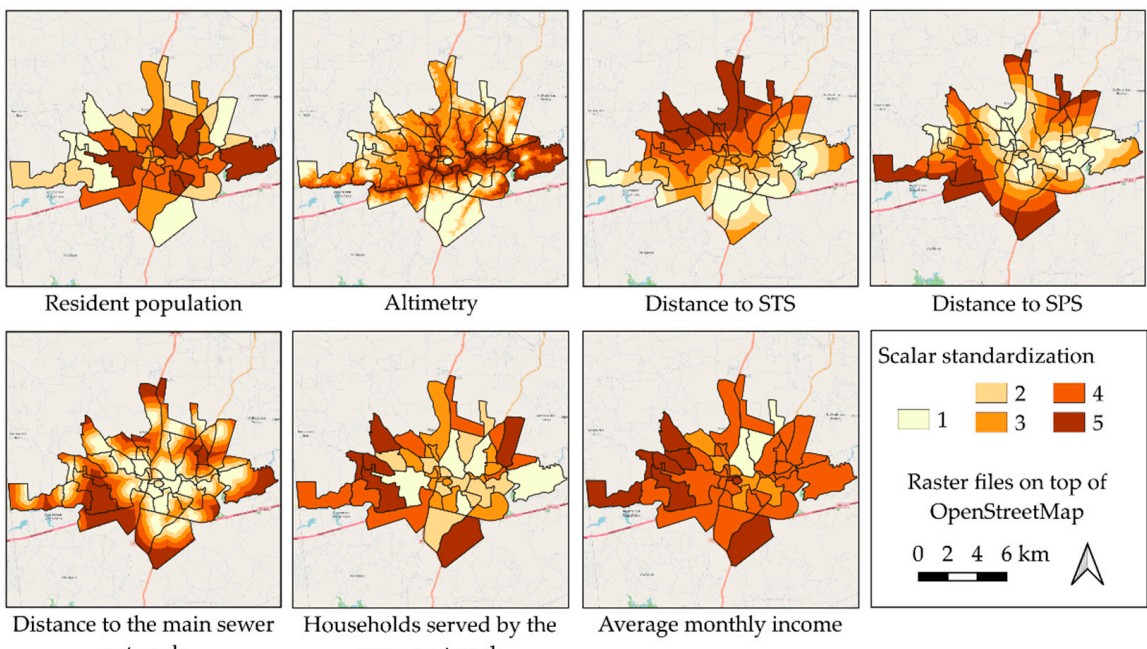

**Figure 5.** Standardization of criteria to assess the degree of insufficiency of the sanitary sewage network, using a scale from 1 to 5.

**Table 3.** Degree of importance of the AHP method criteria.

| Value | Definition | Explanation |
|-------|------------|-------------|
| 1 | Equal importance | Both criteria contribute equally to the objective. |
| 3 | Slightly more important | Experience and judgment show that one criterion is slightly more important than the other. |
| 5 | Moderately moreimportant | Experience shows that one criterion is moderately more important than the other. |
| 7 | Strongly moreimportant | Analysis and experience show that one criterion prevails over the other, and its importance is demonstrated in practice. |
| 9 | Extremely important | With a high degree of certainty, one criterion is absolutely predominant over the other. |
| 2,4,6,8 | Intermediate values | They can be used when looking for a compromise condition between two definitions. |

The degree of relative importance of the criteria was defined by the judgment of individuals with knowledge about water supply and sanitary sewage systems, in the following segments: university professors; engineering masters; city hall servants; sanitation company servants and civil society in general. Similarly, in the study by [23], the profile of the individuals consulted was chosen so that they met three aspects: the follow-up of the periodic water crises through which the area under study passed; in-depth knowledge of the system's technical and operational issues; to develop different functions in Urban Water Supply Systems.

When standardized, the criteria can be combined to prepare the final maps of the risk of water shortages and the degree of insufficiency of sanitary sewage. In this research, the Weighted Linear Combination (WLC) model was used (Equations (1) and (2)), where the final result will be the sum of the values of all criteria multiplied by their respective weights [54,55], since the WLC model is the most frequently used in GIS processes [56]. Using the weights obtained, similar to [19], in the weighting of the criteria, we have:

$$\text{RWS} = \sum_{k=1}^{n} W_k C_k = 0.289C1 + 0.182C2 + 0.118C3 + 0.096C4 + 0.125C5 + 0.085C6 + 0.059C7 + 0.046C8 \quad (1)$$

$$\text{DISS} = \sum_{K=1}^{N} W_k C_k = 0.136C1 + 0.064C2 + 0.159C3 + 0.061C4 + 0.229C5 + 0.196C6 + 0.155C7 \quad (2)$$

where:
RWS = Risk of water shortage
DISS = Degree of insufficiency of sanitary sewage
Wk = Criterion weight k
Ck = Criterion k
n = Number of criteria

## 4. Results

### 4.1. Relative Matrix Importance of the Criteria

After the collection, processing and analysis of the data obtained, and after the matrices of the relative importance of the criteria are obtained, the definition of the average of the scores according to the judgment of each individual consulted can be obtained, which can be observed in Tables 4 and 5 for the RWS and DISS analyses, respectively. During the application of the AHP method and after new considerations were made, a value of 0.089 was reached for the CR of the RWS analysis, and 0.077 was reached for the CR of the DISS analysis. Therefore, the weights obtained from each criterion can be used, since the CR < 10%.

**Table 4.** Relative matrix importance of the criteria, CR and weights obtained for the criteria for determining the risk of water shortages.

| | C1 | C2 | C3 | C4 | C5 | C6 | C7 | C8 | Weight Obtained |
|---|---|---|---|---|---|---|---|---|---|
| C1 | 1 | 3 | 4 | 4 | 3 | 2 | 3 | 4 | 0.289 |
| C2 | 1/3 | 1 | 3 | 3 | 2 | 2 | 3 | 3 | 0.182 |
| C3 | 1/4 | 1/3 | 1 | 2 | 1 | 2 | 4 | 2 | 0.118 |
| C4 | 1/4 | 1/3 | 1/2 | 1 | 1 | 2 | 3 | 2 | 0.096 |
| C5 | 1/3 | 1/2 | 1 | 1 | 1 | 3 | 4 | 2 | 0.125 |
| C6 | 1/2 | 1/2 | 1/2 | 1/2 | 1/3 | 1 | 2 | 3 | 0.085 |
| C7 | 1/3 | 1/3 | 1/4 | 1/3 | 1/4 | 1/2 | 1 | 3 | 0.059 |
| C8 | 1/4 | 1/3 | 1/2 | 1/2 | 1/2 | 1/3 | 1/3 | 1 | 0.046 |
| | Consistency ratio (CR) = 0.089 | | | | | | Total | | 1.000 |

**Table 5.** Relative matrix importance of the criteria, CR and weights obtained to assess the degree of insuf-ficiency of the sanitary sewage network.

| | C1 | C2 | C3 | C4 | C5 | C6 | C7 | Weight Obtained |
|---|---|---|---|---|---|---|---|---|
| C1 | 1 | 4 | 1 | 3 | 1/2 | 1/3 | 1/2 | 0.136 |
| C2 | 1/4 | 1 | 1/2 | 1 | 1/2 | 1/3 | 1/3 | 0.064 |
| C3 | 1 | 2 | 1 | 3 | 1 | 1 | 1 | 0.159 |
| C4 | 1/3 | 1 | 1/3 | 1 | 1/2 | 1/3 | 1/3 | 0.061 |
| C5 | 2 | 2 | 1 | 2 | 1 | 3 | 2 | 0.229 |
| C6 | 3 | 3 | 1 | 3 | 1/3 | 1 | 2 | 0.196 |
| C7 | 2 | 3 | 1 | 3 | 1/2 | 1/2 | 1 | 0.155 |
| | Consistency ratio (CR) = 0.077 | | | | | Total | | 1.000 |

According to the individuals consulted, when it comes to water supply, the criteria that most influence the risk of shortages are C1 (resident population), C2 (altimetry) and C5 (distance to the main network), totaling 59.6% of all weight. The weighting of the distance to the main network can point out the relationship between the expansion of the urban area and the supply network, where residents at the ends of the network are more susceptible to a lack of water. In addition, it can indicate the cause-and-effect relationship between the implementation of infrastructure appropriate for the location and the improvement of the quality of the service.

The criteria that appear as intermediate are C3 (distance to reservoirs), C4 (distance to treated water pumping stations) and C6 (households served by the network). It can be seen that areas with a greater number of households connected to the supply network create a greater demand in that area, and, consequently, with the decrease in supply, households will soon feel the effects of shortages. In addition, it is understood that households that are far from reservoirs and treated water pumping stations will have greater supply difficulties since, in some cases, there is no infrastructure that takes water to these households. In addition, consumption over the long term of the network path can eventually reduce the water supply in points far away from the reservoirs.

The criteria that appear with less relevance in relation to the others are C7 (income) and C8 (distance to WTS). In contrast, [23] reported that income had a more significant influence, and despite the socioeconomic situation providing greater capacity to reserve water to the resident, it is inferred that, due to the proposal to universalize the coverage of public sanitation services, "the contracts for the provision of public sanitation services must define universalization goals that guarantee the service of drinking water to 99% (ninety-nine percent) of the population and sewage collection and treatment to 90% (ninety percent) of the population until 31 December 2033", regardless of the population's income and any other socioeconomic or demographic characteristics. This was highlighted in the new Legal Framework for Basic Sanitation in Brazil, Law 14,026 [57]. Regarding the WTS, whose objective is to treat the water to the point that it is suitable for human consumption, it is understood that, although they have an important function, if the WTS is in adequate operating conditions, it will have less interference in the shortage, being, in this case, less relevant in relation to the other criteria analyzed.

In turn, when the subject is sanitary sewage, the criteria that most influence areas susceptible to a lack or insufficiency of sanitary sewage are C3 (distance to sewage treatment plants), C5 (distance to the main sanitary sewage network) and C6 (households served by the network), adding up to 58.4% of the entire weighting. The distances to the main network and the STS and the lack of sewage connection identify the infrastructure difficulty for the adequate connection of a sanitary sewage system to the referring home, resulting in many cases where the sewage is released directly into the street or the nearest water body. In the same way, without infrastructure equipped with sewage treatment stations, sewage collection becomes impractical, since there is no treatment. As a result, the sewage cannot be released to its final destination. The criteria that appear in the third place as the most important are C1 (resident population) and C7 (income). Although there has been progress toward expanding access to water supply and sanitation services, as highlighted by [58], access to these services by income class is still unequal, and the deficit in the basic sanitation sector is greater, especially with regard to sanitary sewage, in areas where the population with lower incomes is concentrated.

The criteria that appear with less of an influence are C2 (altimetry) and C4 (distance to sewage pumping stations). The SPSs are part of the collected sewage transport system; they lift the collected sewage from lower points to higher points, and they are used in regions where gravity cannot be used in favor of sewage transport. As this is a factor that depends on the relief of each region, as well as the altimetry criterion, it is considered that such criteria have less relevance, in relation to the criteria adopted, to evaluating the DISS in a given area.

### 4.2. Mapping of Areas with Risks of Water Shortages

Observing the vulnerabilities and strengths of the UWSS and SSS, in a second moment, this work proposed the generation of a mapping of areas with risks of water shortages and areas with a degree of insufficiency of sanitary sewage. Thus, to evaluate the most susceptible areas, maps of these two aspects were produced for the entire study area, where it is possible to carry out a spatial assessment and generate visual outputs that help in the understanding of the study and in the decision making by urban managers, as highlighted by [15]. When generating the map, it is shown that the high potential MAR sites are located in the western portion of the study area. Figure 6 shows the mapping of the RWS in the urban area of Caruaru.

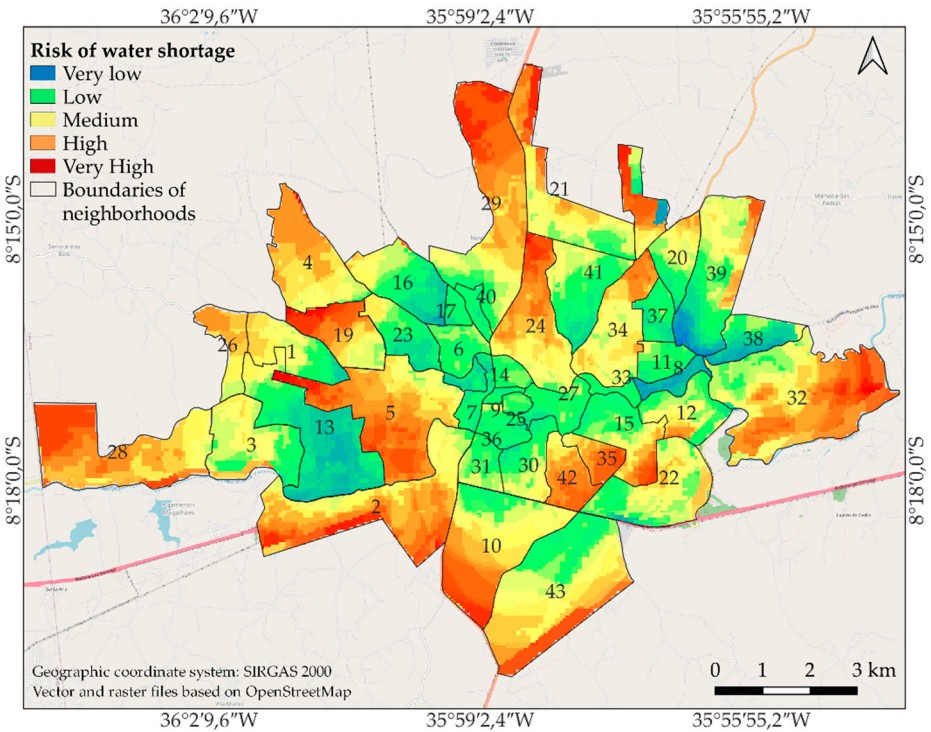

**Figure 6.** Mapping the risk of water shortages for the urban area of Caruaru, Pernambuco.

From Figure 6, it can be seen that the neighborhoods with areas with a "Very high" risk of shortages were: Nossa Senhora das Graças (28); Cidade Alta (10); Agamenon Magalhães (2); Nova Caruaru (29); Kennedy (5); Rendeiras (32); José Carlos de Oliveira (19). The neighborhoods with "High" risk areas were: Nina Liberato (26); Maurício de Nassau (24); Vassoural (42); Salgado (34); Santa Rosa (35); Andorinha (4). It can be seen that the neighborhoods with the highest risk of shortages are located in more peripheral areas of the city (with the exception of Maurício de Nassau). A similar result was found by [23] for the city of Campina Grande-PB. These zones are further away from the distribution reservoirs which create the main water supply network. Among the criteria analyzed, the distances to water supply and altimetry infrastructure had the greatest influence in the Cidade Alta, Agamenon Magalhães, and Nina Liberato neighborhoods. The distances to infrastructure, this time in conjunction with the income criterion, were predominant in the Andorinha, Nossa Senhora das Graças and José Carlos de Oliveira neighborhoods. In the Kennedy, Rendeiras, Maurício de Nassau, Vassoural and Santa Rosa neighborhoods, the preponderant criteria by which to classify these neighborhoods as having a "High" or "Very high" risk of shortages were the resident population and the number of households served by the network.

It is usual for residents of these neighborhoods to complain about the lack of water supply, even in periods where there is no official rotation system by the supply company. It

is also common for residents of neighborhoods with a "Very high" and "High" risk to store water in small reservoirs that, in general, do not meet the needs of users in terms of the volume and quality of water within potability standards. As [59] highlights, even with the existence of recommendations for the proper management of water stored in homes and for the maintenance of water quality, it is observed that, in practice, the recommendations are not always being applied. Despite this, users' predisposition to accumulate water appears to be a relevant condition to mitigating the risks of shortages, especially in long periods of water crisis. In watersheds, changes in the quality of water bodies can be identified through water quality monitoring. However, in general, only the punctual analysis does not allow for a broad view of the behavior of change in the basin. Thus, with the use of a GIS, it is also possible to obtain maps covering water quality parameters.

On the other hand, the neighborhoods with "Low"-risk areas were: São João da Escócia (37); Cidade Jardim (11); Petrópolis (30); Indianópolis (15); Divinópolis (14); Morro Bom Jesus (25); São Francisco (36); Riachão (33); Universitário (41). The areas with a "Very Low" risk were the following neighborhoods: Jardim Boa Vista (16); Distrito Industrial (13); Maria Auxiliadora (23); Serras do Vale (39); São José (38); Cedro (8). Almost all of these areas are located close to the reservoirs, the main water supply network and the water treatment plants (more specifically, the São João da Escócia, Cidade Jardim and Petrópolis neighborhoods). Some neighborhoods still have a relatively low population, although there is constant construction of popular houses, such as in Jardim Boa Vista, Serras do Vale, Cedro, São José and the Distrito Industrial, where, technically, there is no resident, despite the demand from the installed industries, which is a population equivalent. All of the neighborhoods with a "Low" or "Very low" risk are located at lower altitudes, with the exception of only the Morro Bom Jesus neighborhood. However, even if these areas present favorable criteria, there is an accelerated population growth that increases consumption and must be taken into account, as it will eventually overload the water supply system even more.

### 4.3. Mapping of Areas with a Degree of Insufficiency of Sanitary Sewage

Analogously to the RWS map, Figure 7 shows the mapping of the DISS in the urban area of Caruaru, followed by the analysis of the result.

From Figure 7, it can be inferred that the neighborhoods with areas with a "Very high" degree of insufficient sanitation were: Agamenon Magalhães; Luiz Gonzaga (21); Nova Caruaru; Vila do Aeroporto (1). The neighborhoods with areas with a "High" grade were: Andorinha; Kennedy; Jardim Boa Vista; José Carlos de Oliveira; Salgado; São João da Escócia; Rendeiras; Verde (43). The neighborhoods that presented a "Very high" grade of insufficient sanitation are all far from the main sewage network and STS. Those criteria were the main reason for these neighborhoods receiving this degree of insufficiency of sanitary sewage, according to the individuals consulted. In the case of neighborhoods that received a "High" grade, other characteristics are more significant, such as the low number of households served by the sewage system, income and, in the cases of Rendeiras, Kennedy and Salgado, the large resident population. Similar to the water supply service, it is common for residents of these neighborhoods to criticize the lack of adequate sewage, reporting situations of sewage being dumped directly into the streets and canals and, in many cases, burst pipes.

A systematic in situ verification of information on criticisms and complaints regarding water supply and sanitary sewage was not part of the scope of this work, but the lack of water and open sewers in the daily life of a portion of the population is already common knowledge. In addition, there are some articles from radio and TV stations that show the lack of these services, such as "Residents complain about burst sewage in Caruaru neighborhood" [60]. The [61] highlights that "Residents of Caruaru close federal highway to protest lack of water." These and other articles can be seen in the [62,63]. Even more, according to [64], water rationing harms the fight against water COVID-19, primarily in lower-income areas.

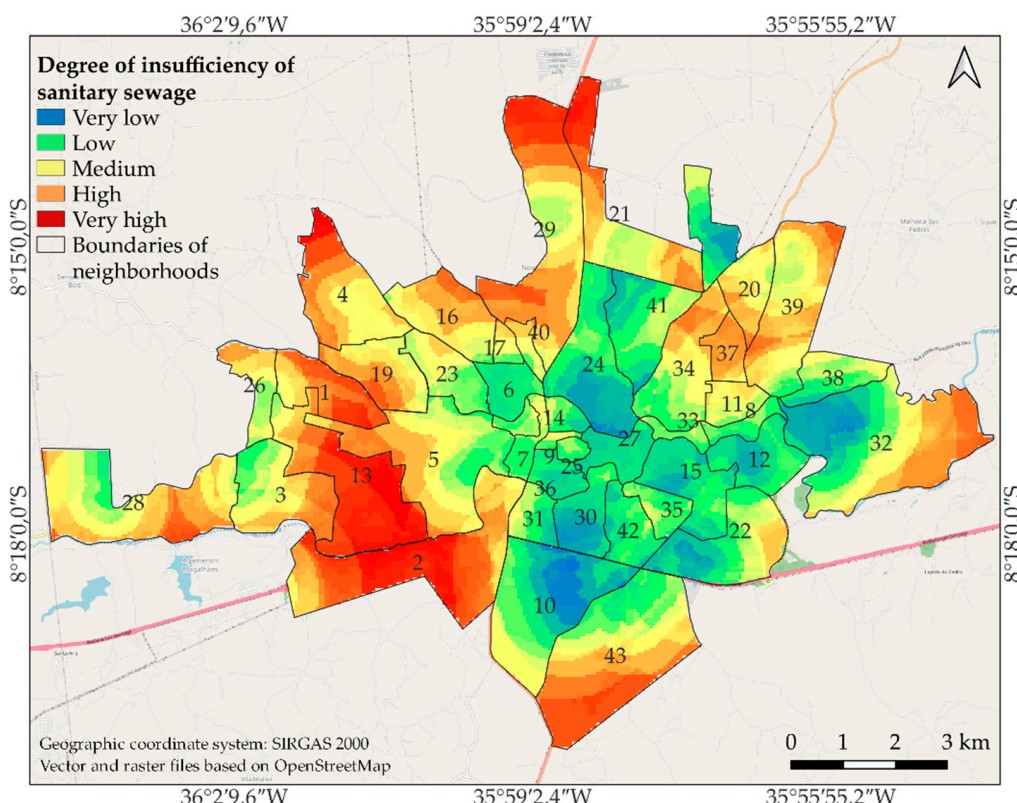

**Figure 7.** Mapping of the degree of insufficiency of the sanitary sewage network for the city of Caruaru, Pernambuco.

On the other hand, the neighborhoods with areas with a low degree of sanitary sewage insufficiency were: Boa Vista (6); João Mota (18); Caiucá (7); Nossa Senhora das Dores (27); Cedro; Universitário. The neighborhoods with areas with a "Very low" degree were: Cidade Alta; Maurício de Nassau; Petrópolis; Indianópolis; Deputado José Antônio Liberato (12). Although there are elements of the sewage system in these neighborhoods, it is worth noting that the lack of maintenance of the collection network causes harm to the system as a whole. Neighborhoods with higher risks may be suggestive areas for the adoption of public policies regarding the water and sewage components of basic sanitation in urban areas.

## 5. Discussion

The results show how RWS and DISS maps can be important tools in urban water management, not only for water management agencies and companies but also for municipal managers. The maps show how water and sewage services are heterogeneously distributed in the urban area with regard to the safety of service. With the population growth and, consequently, the increase in the demand for water in the city of Caruaru, the importance of managing this demand also grows, either through the control and reduction of losses or through the economical and rational use of water by consumers. Users often criticize the lack of water, even in times of non-formal rationing. Regarding loss control, [2] showed that savings of 57 GWh are possible in a period of twelve years when measures are developed that are different from the common ones for the monitoring and control of water losses in intelligent water management. Moreover, for [65], the main actors are public managers, so it is up to governments to create the conditions to solve problems where markets can offer appropriate technologies at an affordable price.

It is worth noting that efficient management should consider a periodic assessment of the UWSS and the SSS. The increase in demand, for example, must be managed together with the entrepreneurs responsible for the new subdivisions, the construction of many of which is underway in 2022 in the neighborhoods of Andorinha, Cidade Alta, Nova

Caruaru, Nina Liberato, Serras do Vale, Verde, Luiz Gonzaga and Universitário, leaving it up to the municipality service provider and other interested parties to determine how the responsibility for better service to the population will be shared.

In view of the growth in demand for water services, in addition to existing springs, there is a search for new sources. For [66], measures considered effective in minimizing the water supply problem in the region in which Caruaru is located, in the coming decades, are the projects for the Integration of the São Francisco River with the Northeastern Watershed and the Agreste Adductor. This, with a budget of approximately BRL 1.2 billion, started in 2013 and was expected to be completed in 2021; however, the work continues in 2022 [67].

In this context, for [68], water supply and sanitation services in urban areas should prioritize their implementation for groups of families, not for individual families. The justification is the advantageous cost reduction that was achieved by the provision for groups of residences, such as the condominium water supply system in the city of Parauapebas, in the state of Pará, in the north of Brazil. According to [69], the condominium network in Parauapebas covered 287 km of streets, with a total length of only 43 km. Considering that a conventional network would have been established along the entire length of the urban road system, there is an 85% reduction in the total length of the pipelines necessary to complete the water supply network, resulting in great savings in the excavation, breaking and resurfacing of sidewalks, fewer materials as well as a smaller and shorter disturbance of the urban population in the execution of the works.

Another project, highlighted by [70], led to the extension of a 1.3 km water distribution network to the southern area of the city of Iquitos, Peru. In total, 1030 households were connected to the water supply system after the installation of a condominium water and sewage system in the Manuel Cardozo Dávila neighborhood. As a result, diarrheal diseases decreased by 37% for children under 5 years of age from 2003 to 2004. From [71], it was stated that the situation regarding Caruaru in relation to the water supply system "requires expansion of the system." Due to water scarcity, the high demand for water resulting from population growth and an improved quality of life and the rotation of the water supply imposed in the city, this situation is very likely to continue. In view of the examples of Parauapebas and Iquitos, it is interesting to note that the neighborhoods that need the expansion of the water supply network, such as Nossa Senhora das Graças, Nina Liberato, Andorinha, Nova Caruaru, Serras do Vale and Verde, are to receive a condominial supply system. Such neighborhoods have a rapid growth in the number of households. In a considerable part of them, the housing arrangement is subject to the implementation of the condominial system of water supply, as well as sanitary sewage. Furthermore, as [72] found, cities should know how to manage not only resource abundance or short-term scarcity but also long-term scarcity. The authors argue that this is the main way to generate water sustainability in urban areas of the future.

Of the forty-three existing neighborhoods in Caruaru, twelve had a "Very high" or "High" degree of insufficient sanitation, thus pointing to more effective actions regarding the SSS in both qualitative and quantitative terms. In these neighborhoods, the relatively long distances to the devices that make up the SSS stand out, as well as the low number of households served by the network. Thus, it is expected that the system will be implemented in these areas and the sewage will be collected and treated in the STS of the municipality. It is worth noting that sewage must be treated in order to meet the quality standards of the receiving body, and the sludge generated must be properly disposed of in a licensed landfill. The municipality has two landfills: the Caruaru Sanitary Landfill and the Caruaru Waste Treatment Center. In addition, ensuring channels of communication with society and promoting continued actions in environmental education are essential interventions for the greater effectiveness of the service.

Another important aspect is the areas with residents with lower average incomes, such as the Andorinha and José Carlos de Oliveira neighborhoods. To serve them, it may be essential to adopt social programs such as the Pró-Conexão Program, developed in 2011 to serve customers without the economic means to connect their sanitary facilities to

Sabesp's networks. Carried out in partnership with the Government of the State of São Paulo, the initiative was aimed at families who receive up to three minimum wages per month and sought to prevent the irregular disposal of sewage. As part of the program, the costs of installing the interconnections are fully paid by the government, which pays 80% of the work, and by Sabesp, which pays 20%. In the neighborhoods to be served, residents of the locality go to the houses to explain the advantages of connecting to the collection network [73], a characteristic of smart cities, where citizens capable of participating in municipal initiatives are needed.

When it comes to SSS and UWSS, it is known that such systems influence the lives of the population collectively; therefore, the participation of all interested parties and the support of local authorities and the federal government are essential. The projects carried out in Sub-Saharan Africa to upgrade and expand existing infrastructure, for example, reaffirm the importance of stakeholder participation for the success of Integrated Urban Water Management (IUWM) initiatives [74]. It is also important that the municipal, state and federal institutions develop IUWM projects in a strategic and integrated way.

In this context, some indexes and activities can be considered and analyzed for the periodic evaluation of the UWSS and the SSS, seeking a more efficient management, such as: macro measurement index; average per capita water consumption; distribution loss index; treated sewage index referring to the water consumed; urban water service index; urban sewage service index, collected sewage treatment index; loss rate per connection. In addition, as already mentioned, the introduction of environmental education in the daily life of the population is fundamental for the preservation of water resources.

The work being done in Sub-Saharan Africa shows some of the benefits of integrating stakeholder participation from the beginning of the project. In Cape Verde, for example, community leaders and the population identified the need for hotels and similar establishments to have rainwater reuse systems, mainly considering the reuse of water from baths and washbasins. Therefore, it was necessary to hold awareness and environmental education lectures with their customers and employees [75]. From a technical and social point of view, such measures should be discussed and analyzed in semi-arid cities such as Caruaru. Furthermore, the results of this research—similar to those of [76], who presented how standardized climate indices for monitoring systems can be useful in terms of preparing for drought episodes, and [17,77], who pointed out that integrated MCDA and geospatial techniques provide a valuable resource for the rainwater harvesting strategy—show that it is possible, with pertinent information and sound decisions, to have a greater capacity to adapt to and prevent adverse situations in semiarid regions with regard to urban water use.

## 6. Conclusions

The results of this research, together with the literature review, showed that the management of water supply and sewage services in Caruaru must be based on a spatial format that integrates non-structural and structural aspects, as well as governmental, private and societal actions that affect civilians as a whole.

It was found that, through the basic diagnosis of the UWSS and the SSS of Caruaru, the distributed and macro-measured volume for the population of the urban area of the municipality in the year 2021 was 20,611,914 m3, an average of 156.00 L of water per inhabitant per day. This is a value considered within the ideal standard; however, part of this value is lost along the way due to system failures, mainly due to leaks in the pipes.

In turn, with the RWS and DISS mapping, it was observed that the areas that presented "Very high" and "High" risks of water shortages and insufficient sanitary sewage in the generated maps were those located in more peripheral areas of the city, especially in the North and West. These areas can act as indicators for decision making and more effective actions in the management and planning of water and sewage systems.

It was seen that the criteria of the resident population and the distance from the main network (water or sewage) had a greater impact with regard to the risk of water shortages and the insufficiency of sanitary sewage through the conventional network. Other criteria that drew attention to a new discussion were, respectively, altimetry and distance from sewage treatment plants. Thus, the planning, implementation and expansion of the UWSS, the SSS and the urban area of Caruaru and other municipalities must be rethought to integrate with each other more efficiently.

In addition, it was also seen that the use of GIS was relevant, since the results generated show areas that are most lacking in terms of water and sewage services, and, thus, risk minimization plans and goals can be devised for these areas with greater urgency.

It was noted that, despite projects and construction in progress that implement a greater supply of water in the region, such as the Agreste Aductor, it is necessary to consider that, according to the trends of population studies, the population of Caruaru will continue to grow at a faster rate than most of the other cities in the region, which causes a greater demand for water than the available supply.

Thus, it is concluded that the use of high-resolution spatial databases for the planning of urban services, as carried out in the present work, provides a greater level of confidence for solutions that can be implemented in the expansion of service networks to the population.

It is also concluded that, to avoid searching for water from sources that are increasingly distant from the point to be supplied, actions to capture alternative sources and measures of conscious water use must be implemented as technologies to support the urban water cycle in Caruaru and other cities in semi-arid regions.

Considering that this research observed, in its evaluation, the basic sanitation in the dimensions of water supply and sanitary sewage, the proposed model can have its application extended to municipalities in similar regions and to municipalities in regions with other characteristics, and it can also be included in urban solid waste management and/or urban drainage services in the GIS-MCDA assessment.

Finally, it was possible to achieve the proposed objectives, and it is expected that the results of this research can contribute to: enhancing the development of the infrastructure of conventional systems; using maps of risk areas to support planning; verifying the use of rainwater and ash urban supply; facing the reduction of water losses; managing the supply of the adequate collection, transport and treatment of sanitary sewage in the integrated water cycle in the urban environment.

**Author Contributions:** Conceptualization, M.C.d.O.S. and J.A.C.; data curation, M.C.d.O.S.; formal analysis, M.C.d.O.S.; funding acquisition, R.S.V.; investigation, M.C.d.O.S.; methodology, M.C.d.O.S.; project administration, J.A.C.; resources, J.A.C.; supervision, R.S.V. and J.A.C.; validation, R.S.V. and J.A.C.; writing - original draft, M.C.d.O.S.; writing - review & editing, R.S.V. All authors have read and agreed to the published version of the manuscript.

**Funding:** Capes project n. 88887.163498/2018-00.

**Data Availability Statement:** The data presented in this study are available on request from the corresponding authors.

**Acknowledgments:** The authors are grateful to Fundação de Amparo à Ciência e Tecnologia de Pernambuco (FACEPE) and Capes project n. 88887.163498/2018-00 for funding and a postdoctoralfellowship. The authors are also grateful to the Brazilian Institute of Geography and Statistics (IBGE), Companhia Pernambucana de Saneamento (Compesa) and the City Hall of Caruaru for their availability and urbanity and the information provided, which were fundamental for the development of this article.

**Conflicts of Interest:** The authors declare no conflict of interest.

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
