# Peer review of "Risk Mapping of Water Supply and Sanitary Sewage Systems in a City in the Brazilian Semi-Arid Region Using GIS-MCDA"

_water, doi:10.3390/w14203251_

Round 1

Reviewer 1 Report

The authors have attempted to present the risk mapping of water supply and Sanitary sewage systems in a city in the Brazilian semi-arid region using the GIS-MCDA tool. Multi-criteria decision analysis is a widely used criterion that involves many environmental parameters for the risk or hazard or potential site estimation. However, the manuscript does not clearly illustrate the environmental parameters taken for such analysis. I also wonder, how the authors have used two main components in this study 1) water shortage and sanitary sewage system. Both are different entities and how it has been correlated? No statistical analysis has been carried out to find the significance level in the present study. The time of data and its frequency is not mentioned. English of the manuscript needs to be revised throughout the manuscript for better understanding and clarity. In addition, that sentence needs to be restructured to maintain the manuscript flow for better understanding.

Major comments:

  1. page 1, lines 11-27; Abstract tells about the source of the data but it doesn't talk about the type of the data, its frequency, and duration. Without any result interpretation in the abstract, it seems to be superficial. Please rewrite the whole abstract indicating your findings.
  2. page 2, lines 40-47; You can start from According to Soares et al., (2021)......
  3. page 2, line 48; There are ... such concerns. Please delete this line.
  4. page 2, lines 53-55: Please rephrase the sentence for better understanding.
  5. page 3, lines 125-126: Please rephrase the sentence.
  6. page 3, lines 137-139: Please rephrase the sentence. Please clarify.
  7. Page 4, line 187-188: Please delete: In view of the above.......... context of smart cities.
  8. Page 4: The objective should be very clear which is missing presently. Moreover, the introduction is very lengthy and needs to be very precise. Please keep only relevant statements which are matching with your objective and delete the general statement from the introduction.
  9. page 4, lines 194-195; The average annual rainfall is about 750mm, however, the evaporation rate is 2500mm/year. With this rate, no groundwater will be left. How you can justify this?    
  10. page 5, please concise the study area description. Please remove irrelevant statements.  
  11. Figure 1 and figure can be combined for better clarity. Table 1 data can also be displayed in figure1. Even figure 4 information can be easily integrated with figure 1.
  12. Figure 3: what is the relevance of this figure? Please delete.
  13. Table 2; Source of data is mentioned but there is no clarity regarding the period of data, the scale of the data, and its relevance in developing the model. Rainfall is an important parameter along with the evaporation of water and ground soil quality but these parameters are not addressed in the manuscript.
  14. Page 10, line 327-329; How the criteria were selected. Please provide a standard reference for doing so.
  15. Page 10, line 357; r.class. Please correct it.
  16. Figures 5 & 6; To run the model, how many samples were (point shapefile) used to interpolate the data? Please explain the meaning of standardization on a 1 to 5 scale in the map itself.
  17. Please explain the importance of figure 1 and figure 2. How the sanitary network is related to water scarcity? Please explain.
  18. What is a modeling error?
  19. Table 4: Criteria selected for water shortages and insufficiency of the sanitary sewage network are the same. Please explain the relevance of the insufficiency of the sanitary sewage network to the water shortage.
  20. Page 13, line 422-424; According to the .... totaling 59.6% of all weight. Authors are advised to justify how without considering rainfall data, is this analysis appropriate for deciding on the water shortage model?
  21. Figures 7 and 8 need to be explained very precisely with their importance and their correlation in the present study.
  22. Results and discussion should be rewritten based on the correction suggested.
  23. The conclusion also needs to be revised completely.         

Reviewer 2 Report

The article concerns the risk mapping of some municipal systems. It may be interesting for readers of Water. The following requests/suggestions should be taken into account to improve the quality of the manuscript.

-  In my opinion it is not clear what is the novelty of this study. Why do you contribute to the progress of science? Please explain this issue.

-  In environmental engineering (as in most engineering applications) the basic risk definition applies, which presents risk as a product of the probability of undesirable events occurrence and losses resulting from it. Please explain why you are not using this well-known methodology.

-  In the water shortage risk analysis, the hydraulics of the water supply system cannot be ignored, i.e. pipe diameters, pressure distribution, flow. It is also very important to take into account the failure frequency and technical condition of the water supply network. I consider not taking these parameters into account as a mistake.

-  What is the basis of the Criterion weight adopted in formulas 1 and 2?

-  The conclusions are general. What is the new result of the analysis performed? In my opinion, it is obvious that the greatest risk of shortages will be in the peripheral areas of the city.

Reviewer 3 Report

The research deals with important issue of the risk mapping of water supply and sanitary sewage systems in a city in the Brazilian semi-arid region using GIS-MCDA. The secondary data used in this research were collected from IBGE, Compesa, and the City Hall of Caruaru and were processed using the QGIS 3.12 BucareÅŸti software. The Pernambuco Tridimensional database and the Analytical Hierarchy Process method were used in the process to generate the maps. Comments: The criteria were selected based on the similarity with the criteria for assessing the risk of water shortages and on the expertise of specialists in the sanitation area.

Methods need theoretical background. There are many MCDM tools ranging from fuzzy to simple one. Why AHP was selected? What is the value added with respect to existing methods? The relevance of the proposed methodology should be definitely discussed further. Concerning the methodology, although the structure of the AHP is rather clear, there are not enough information on the process to develop it. Particularly, since the authors state that the use of expert knowledge is a relevant point of their work, I believe that further details are needed on this. Eg. line 398: According to the typographic rules, the decimal part is separated by a dot, and the comma is used, for example, to separate thousands, so the whole text, figures, tables should be prepared according to this rule. More specifically, which experts were involved? Did you find any differences and/or inconsistencies in their belief? Did you find relevant literature identifying the decision-makers’ weight for indicators to be considered in the analysis? Please justify the choice of the criteria considered for mapping areas at risk of water shortages.

Round 2

Reviewer 2 Report

Thank you for your answers. The current version of the article is proper to be considered for publication. 

Author Response

According to Reviewer 2: - The current version of the article is proper to be considered for publication.

However, it still suggests an improvement in the conclusions. Which were reviewed.

Reviewer 3 Report

The research deals with important issue of the risk mapping of water supply and sanitary sewage systems in a city in the Brazilian semi-arid region using GIS-MCDA. The secondary data used in this research were collected from IBGE, Compesa, and the City Hall of Caruaru and were processed using the QGIS 3.12 BucareÅŸti software. The Pernambuco Tridimensional database and the Analytical Hierarchy Process method were used in the process to generate the maps. Comments: Remarks: Line 114: The choice of reference should be supplemented with respect to the information system based the multi-criteria decision analysis as the preferred approach in several researches because it involves a combination of multiple criteria in a weighted way and also produces visual results, important for decisions in the urban environment, being used to: decide the multi-criteria support for water management; optimize the layout of water supply pipeline systems; do mapping of water scarcity risks (e.g. Ref. Modelling water distribution network failures and deterioration, 2017, IEEE International Conference on Industrial Engineering and Engineering Management 2017-December, 924-928. DOI 10.1109/IEEM.2017.8290027; Valis, D. Perspective renewal model for water distributions systems, 26th Conference on European Safety and Reliability (ESREL), 2017, Risk, Reliability and Safety: Innovating Theory and Practice, 1050-1055. According to the typographic rules, the decimal part is separated by a dot, and the comma is used, for example, to separate thousands, so the whole text, figures, tables should be prepared according to this rule. Eg. line 355, in theTable 5. Relative matrix importance of the criteria, CR and weights obtained to assess the degree of insufficiency of the sanitary sewage network. Give some details about the efficient management, that should be considered as a periodic assessment of the UWSS and the SSS. What adjustable impact factors do you want to take into account in investigation the water quality dynamics and its impact mechanism. Please include this in the text. Include some information in the conclusions abou future perspectives of the work.

Author Response

The typography of tables 4 and 5 has been adjusted, as well as the entire text. Some indexes and activities can be considered for periodic evaluation of the UWSS and the SSS, seeking a more efficient management. The use of geoprocessing can also be a tool for integrating and analyzing water quality information. As a future perspective, the proposed model may also include urban solid waste management services and/or urban drainage. These recommendations were inserted into the manuscript in more detail.